# Estradiol Removal by Adsorptive Coating of a Microfiltration Membrane

**DOI:** 10.3390/membranes11020099

**Published:** 2021-01-30

**Authors:** Zahra Niavarani, Daniel Breite, Andrea Prager, Bernd Abel, Agnes Schulze

**Affiliations:** 1Leibniz Institute of Surface Engineering (IOM), Permoserstraße 15, D-04318 Leipzig, Germany; zahra.niavarani@iom-leipzig.de (Z.N.); Daniel.Breite@iom-leipzig.de (D.B.); andrea.prager@iom-leipzig.de (A.P.); bernd.abel@iom-leipzig.de (B.A.); 2Wilhelm-Ostwald-Institute of Physical and Theoretical Chemistry, Universität Leipzig, Linne-Strasse 2, 04103 Leipzig, Germany

**Keywords:** surface functionalization, interfacial reaction, adsorptive membrane, estradiol adsorption

## Abstract

This work demonstrates the enhancement of the adsorption properties of polyethersulfone (PES) microfiltration membranes for 17β-estradiol (E2) from water. This compound represents a highly potent endocrine-disrupting chemical (EDC). The PES membranes were modified with a hydrophilic coating functionalized by amide groups. The modification was performed by the interfacial reaction between hexamethylenediamine (HMD) or piperazine (PIP) as the amine monomer and trimesoyl chloride (TMC) or adipoyl chloride (ADC) as the acid monomer on the surface of the membrane using electron beam irradiation. The modified membranes and the untreated PES membrane were characterized by scanning electron microscopy (SEM), X-ray photoelectron spectroscopy (XPS), water permeance measurements, water contact angle measurements, and adsorption experiments. Furthermore, the effects of simultaneous changes in four modification parameters: amine monomer types (HMD or PIP), acid monomer types (TMC or ADC), irradiation dosage (150 or 200 kGy), and the addition of toluene as a swelling agent, on the E2 adsorption capacity were investigated. The results showed that the adsorption capacities of modified PES membranes toward E2 are >60%, while the unmodified PES membrane had an adsorption capacity up to 30% for E2 under similar experimental conditions, i.e., an enhancement of a factor of 2. Next to the superior adsorption properties, the modified PES membranes maintain high water permeability and no pore blockage was observed. The highlighted results pave the way to develop efficient low-cost, stable, and high-performance adsorber membranes.

## 1. Introduction

Humans and aquatic species are frequently exposed through the water to substances that cause disruption of the endocrine system, called endocrine-disrupting chemicals (EDC). This exposure has become a serious environmental and health problem worldwide [1,2]. Among these EDCs, natural estrogens (e.g., estrone (E1) and 17β-estradiol (E2)), as well as synthetic estrogen (17α-ethinylestradiol (EE2)), have been receiving increased attention as a class of emerging contaminants due to their high occurrence and persistence in the sewage treatment plants (STP) effluents and receiving natural waters [3,4,5]. Previous research studies have established a possible link between human exposure to estrogenic EDCs and decreasing male sperm counts and increases in several types of cancer [6,7,8]. The decline of fertilization rate and the alteration of the development and reproductive performances of fish and aquatic invertebrates have also been reported [9]. With the concerns regarding the high spread of estrogenic EDCs in water, the European Union has recently introduced a watch list mechanism to monitor the hormones E2 and EE2 amongst other substances to establish future standards for STP effluents discharge of estrogenic EDCs and pharmaceuticals as a part of the European Priority Substances Directive [10]. Nevertheless, concentrations of estrogenic hormones between 0.1 and 10 ng/L have been reported in domestic wastewater effluents and receiving natural water bodies in various cases around the world [3,11,12,13,14]. Hence, developing an effective method for extracting EDCs from water is of vital importance.

Various methods such as catalytic degradation, photocatalytic degradation, biodegradation, advanced oxidation, liquid–liquid extraction (LLE), and ozone reactive LLE have been explored for the removal of estrogenic EDCs from water [15,16,17,18,19,20,21]. In comparison to the mentioned techniques, adsorption was addressed as a more efficient and effective method. In fact, adsorption is an environmentally friendly method that is at the same time efficient and easily accessible. Various adsorbents, e.g., granular activated carbon, chitin, chitosan, ion exchange resin, and carbon-based adsorbents made of industrial and agricultural waste are able to remove E2 from wastewater [22]. Yoon et al. [23] have applied different kinds of powder activated carbon for the removal of E2. Tagliavini et al. [24] studied the adsorption of steroid micropollutants on polymer-based spherical activated carbons. Other sorbents including single-walled carbon nanotubes and multi-walled carbon nanotubes have also shown good performance to remove E2 from aqueous systems [25,26,27,28]. However, high production and regeneration costs make these methods inefficient in water treatment and purification processes. It is, therefore, apparent that new adsorbents for removing estrogenic EDCs from water are necessitated.

Membrane technology, including microfiltration (MF), nanofiltration (NF), and reverse osmosis (RO), are considered viable technologies for the removal of EDCs including natural hormones from water. Past studies suggested that the rejection would be largely controlled by adsorption of hormones to the membrane [29,30,31,32,33,34]. Adsorption leads to the removal of hormones at a much higher level than would be expected based on the hormone’s molecular size. The NF/RO membranes are predominantly prepared as thin-film composite (TFC) membranes. The TFCs are made of a thin polyamide (PA) active layer on the top surface of the membrane. In addition, it has a polyester (PET) backing layer on the bottom, while a polysulfone (PSf) or polyethersulfone (PES) support layer is between the top and bottom layers. Generally, the PET backing layer contributes slightly to the adsorption of EDCs. In contrast, there is still a debate on which of the other two layers play the predominant role in the adsorption of EDCs. It has been discussed in various studies that the main mechanism in the adsorption of hormones, e.g., E2 (containing both hydroxyl and carbonyl groups), is the formation of hydrogen bonds on the active polyamide layer of TFC membranes [31,33,35,36]. Others have postulated that the adsorption might be due to the hydrophobic interaction between hormones (log K_OW_ of E2 = 4.1) and the membrane surface [32].

Steinle-Darling et al. [37] clarified that adsorption of fluoxetine on the PA-PSf layer is higher than that on a commercial PSf membrane. It indicates that the PA layer has a high affinity toward the hormone. In addition, Semiao and Schaefer [38] conducted diffusion cell experiments where the PA-PSf layer separated two similar hormone solutions in a way that the PA and PSf occurred in the opposite directions. The authors reported that the PA layer had shown higher adsorption capacity than the PSf layer in adsorption of hormones from water. PA modification membranes were also studied by Han et al. [34]. They showed that the strong compound binding affinity originates from the hydrogen bonding between PA amide groups and proton donating groups on target compound molecules.

On the other hand, Liu et al. [39] studied the adsorption kinetics of the PA layer by isolating the active PA layer of NF and RO membranes. They peeled off the PET backing layers and dissolved the PSf support layers and argued that the presence of the PSf layer had important impacts on the adsorption capacities and the time necessary to reach adsorption equilibrium. The authors suggested that the EDCs of different physicochemical properties had distinct adsorbed amounts on the two membranes in almost the same order, which mainly resulted from electrostatic attraction/repulsion and hydrophobic interactions.

McCallum et al. [27] applied intermediate stage products such as membrane without polyamide coating layer to carry out some batch and filtration experiments for removal of E2. They observed that the adsorption and desorption of E2 took place at the polysulfone support layer rather than at the polyamide active layer. They mentioned that this behavior is probably due to the hydrophobic interactions.

The published works reviewed above are not in agreement with adsorption mechanism of EDCs on the membranes. It can be said that the adsorption of various EDCs probably takes place at different locations of the membranes (both the membrane surface and inside the pores) [40,41]. It was previously proposed by Semiao and Schaefer that the surface properties of the membrane PA layer and pore size of the membrane have an important influence on the adsorption of the hormones [38]. Accordingly, it has been discussed that a membrane having larger porosity provides easier access to the adsorption sites inside the membrane, i.e., higher adsorption capacity compared to a membrane with smaller pore sizes [29,42].

However, the low water permeation of TFC and PA membranes makes them undesirable for water treatment processes at lower pressure. Additionally, the dense PA layer on the membrane surface leads to blockage of the membrane and lowers the access of the hormones to the adsorption sites and has to be improved.

In this work, we report a novel and efficient method to prepare an adsorber membrane by creating an amide functional coating on porous microfiltration (0.45 µm) PES membrane surface using the concept of interfacial polymerization reaction [43]. The porous support membrane provides the mechanical stability required for operating under high permeation rates. The amide coating was fabricated by means of interfacial reaction between hexamethylene diamine (HMD) or piperazine (PIP) as the amine monomer and trimesoyl chloride (TMC) or adipoyl chloride (ADC) as the acid monomer on the surface of the PES membrane. Electron beam (EB) irradiation was used to immobilize the amine monomers on the surface of the PES membrane via a grafting-to reaction. A subsequent reaction with an acid monomer resulted in the amide coating.

Toluene was added as a swelling agent to increase the surface area. The modification with the amide functionalities did not block the PES membrane pores and increased the E2 uptake without creating any defects or agglomerates.

E2 was chosen as the target molecule due to its high estrogenic potency and common presence in STP effluents [44]. The prepared modified membranes were characterized by means of scanning electron microscopy (SEM), X-ray photoelectron spectroscopy (XPS), Fourier transform infrared spectroscopy (FTIR), water contact angle measurement, water permeance, and E2 adsorption studies.

The enhanced adsorption performance of the modified membranes toward E2 was attributed to the modifications with the functional groups of the membrane surface.

We found an interesting combined effect on the E2 adsorption capacity of the membranes after simultaneous changes of the modification parameters. Four important modification parameters, namely amine monomer type, acid monomer type, EB irradiation dosage, and addition of toluene, were discussed as influential factors.

## 2. Materials and Methods

### 2.1. Chemicals and Materials

Microporous PES (0.45 µm, Millipore Express) and n-hexane were purchased from MERCK (Darmstadt, Germany). Trimesoyl chloride (TMC), hexamethylenediamine (HMD), adipoyl chloride (ADC), piperazine (PIP), and 17β-Estradiol (E2) were obtained from Sigma Aldrich (St. Louis, MO, USA). Ethanol and toluene were purchased from VWR (Radnor, PA, USA). Deionized water in Millipore® quality was used for all steps. All materials were used as they were received from suppliers.

### 2.2. Membrane Modification

In this work, different types of amide modifications were created on the surface of the PES membrane. Figure 1 illustrates the functionalization of the PES membrane by interfacial reaction.

In brief, a PES membrane disk (47 mm diameter) was soaked in an aqueous solution containing the amine monomer (HMD or PIP, 2 wt.%) for 30 min followed by EB irradiation with a dosage of 150 or 200 kGy. The irradiation was performed by means of a home-made electron accelerator (10 mA, 160 kV) under N_2_ atmosphere with O_2_ quantities less than 10 ppm. Afterward, the amine immobilized membranes were rinsed with deionized water three times for 30 min and subsequently dried at room temperature for 60 min. Toluene was added to half of the pre-modified membranes at this stage to investigate the swelling effect. The amine immobilized membranes were immersed in TMC or ADC in n-hexane solution (0.2 wt.%) for 2 min, where the interfacial reaction took place. All modified membranes were dried for 30 min to remove the n-hexane. Then, the membranes were rinsed three times with deionized water for 30 min. Finally, all membranes were dried at room temperature overnight.

The concentration of monomers, the respective irradiation dosage, and the amount of toluene are listed in Table 1. The modified membranes will hereafter be referred to PA-1 to 16.

### 2.3. Membrane Characterization

#### 2.3.1. Water Permeance

A stainless steel filtration cell (16249, Sartorius Stedim Biotech, Göttingen, Germany) was applied to run filtration tests. Water permeance was then calculated by the results of the filtration tests. The permeation time for 50 mL of deionized water was recorded at the pressure of 1 bar. Permeation time was measured for five individual samples and an average of the trials was calculated. Water permeance J was calculated by Equation (1).
(1)$$J=\frac{V}{t\cdot A\cdot p}$$
where *V* is the volume of water passing through the membrane, *t* denotes the permeation time of the water through the membrane, *A* addresses the active surface area, and *p* is the applied pressure. The bubble point of wet membranes was determined by continuously increasing pressure to the point at which the first stream of bubbles emerges.

#### 2.3.2. X-ray Photoelectron Spectroscopy

The chemical compositions of the untreated and modified membranes were investigated by X-ray photoelectron spectroscopy (XPS, Kratos Axis Ultra, Kratos Analytical Ltd., Manchester, UK).

#### 2.3.3. Scanning Electron Microscopy

The morphologies of the modified and untreated PES membranes were studied by scanning electron microscopy (SEM, Ultra 55, Carl Zeiss Microscopy GmbH, Oberkochen, Germany). Magnification ranged from 300- to 25,000-fold. The samples were cut manually and coated with a thin (30 nm) chromium layer by means of the Z400 sputtering system (Leybold, Hanau, Germany).

#### 2.3.4. Water Contact Angle

The surface wettability of the modified and untreated membranes with water was investigated by a static contact angle measurement system (DSA 30E, Krüss, Hamburg, Germany) and the sessile drop method. An average of at least five different sample points was reported.

#### 2.3.5. Adsorption of E2 on Modified Membranes

The adsorption capacities of the modified and untreated PES membranes were measured in a series of batch experiments. In brief, a stock solution of estradiol in ethanol with a concentration of 10 mg∙mL^−1^ was prepared by adding 100 mg of estradiol in a 10 mL volumetric flask and making up to 10 mL with absolute ethanol. The stock solution was sonicated for 15 min. Fifty microliters of this solution was transferred to a 100 mL volumetric flask using an Eppendorf pipette and diluted by 100 mL with an aqueous ethanol solution (10% by volume) and sonicated for 15 min. Finally, an estradiol stock solution with a concentration of 5 mg∙L^−1^ was obtained.

Ten-millimeter pieces of modified and untreated membrane disk samples were placed in a 48-well microtiter plate. To each sample, 200 µL of the aqueous E2 solution with an initial concentration of 5 mg∙L^−1^ were added. The plates were shaken for 30 min at ambient temperature. The supernatant solution was collected and transferred to a new microtiter plate. The final concentration of E2 was measured by means of fluorescent detection (Infinite M200, Tecan, Germany) at an excitation wavelength of 273 nm and an emission wavelength of 305 nm.

The adsorption capacities of the modified membranes were calculated by Equation (2), where *C_0_* is the initial concentration of E2 and *C_f_* is the final concentration after reaching the Equation equilibrium. An average of 5 individual experiments was calculated and reported.
(2)$$\mathrm{Adsorbed}\;E2\;(\%)=\left(\frac{C_{0}-C_{f}}{C_{0}}\right)\times 100$$

## 3. Results and Discussion

The main purpose of this work is to enhance the E2 adsorption on the modified PES membranes, with improving the surface hydrophilicity of the membranes. At the same time, pore-blocking needs to be prevented during the modification reaction of the membranes. Then, the modified membranes were characterized by various techniques for determining the hydrophilicity, pore structure, and chemical composition. Finally, E2 removal experiments were carried out to evaluate the adsorption performance of the modified membranes for the removal of E2 from water.

### 3.1. Membrane Characterization

#### 3.1.1. Water Contact Angle

Water contact angle (WCA) analysis was performed to investigate the surface wettability of the polymer membranes. The water contact angle values for the different modifications and the untreated PES (referred to as REF) membrane are presented in Figure 2. The untreated PES membrane exhibits a hydrophilic surface with a water contact angle of 44°. The modification with the thin PA coating resulted in a moderate decrease of the water contact angles in the range of 37–43°. The lowest water contact angle was observed after modification with PIP and TMC and adding toluene (PA-4). The decrease in the contact angles reveals the enhancement of the wettability of the PES membrane after modification. The reason for this finding could be attributed to increased hydrophilicity due to the presence of hydrophilic amide units in the coating. It is assumed that the decrease in the contact angles discloses the successful formation of the thin amide coating on the surface of the PES membrane. Appendix A shows the experiment for determining the surface wettability of PA-4. The effect of toluene on wettability was also investigated. No significant effect of adding toluene on the wettability of the untreated PES membrane was observed. The water contact angles are listed in Appendix A.

#### 3.1.2. Water Permeance

Membrane performance in terms of permeance was determined by measuring the pure water permeability. The permeance values for the untreated PES membrane and the different modifications are summarized in Figure 3. The untreated PES membrane is already hydrophilic and has a permeance value of 40.1 mL∙min^−1^∙cm^−2^∙bar^−1^. All the PA modifications showed a slight increase in performance with an average permeance value of 41 mL∙min^−1^∙cm^−2^∙bar^−1^. PA-5 with modification parameters of PIP-ADC-150 kGy and without the addition of toluene showed the highest enhancement in permeance with a value of 42.5 mL∙min^−1^∙cm^−2^∙bar^−1^. The slight increase in water permeability can be attributed to the enhanced wettability of the membrane surface. It is assumed that the enhanced wettability of the surface results in the formation of a thin water film on the top of the polymer membrane. This water film helps to prevent hydrophobic interactions and can increase water permeability. Khorshidi et al. [45] reported an average water flux of 7–68 L∙m^−2^∙h^−1^ at a trans-membrane pressure of 1.52 MPa (equivalent to 0.001–0.01 mL∙min^−1^∙cm^−2^∙bar^−1^) for thin-film composite polyamide coated on PES (0.2 µm) microfiltration membrane. A comparison between the permeance values obtained here with what was reported by Khorshidi et al. discloses that immobilizing the amine component by electron beam and the subsequent reaction with the acid reagent could be a better approach to maintain the high water permeability of the PES support. The values from water permeance experiments are presented in Appendix A.

#### 3.1.3. Membrane Pore Structure

The morphology and pore structure of the untreated and modified PES membranes were investigated by SEM. A comparison of SEM images from the surface of the untreated PES and some selected modified membranes can be found in Figure 4. Please note that SEM images of the top surface and cross-section of the modified and reference PES membrane are illustrated in Appendix A. As it could be expected from the water permeation experiments, no pore blockage was observed upon modification with the amide layer. It is observed that the modification does not adversely affect the morphology and no defects could be detected. Thus, the stability of the base membrane is not affected. This means that the modification is an appropriate approach to functionalize the PES membrane with the amide coating without altering the physical structure of the supporting membrane. The SEM results also revealed that this amide modification on the PES membrane is very thin and cannot be detected by SEM.

#### 3.1.4. Membrane Chemical Composition

XPS analysis was carried out to prove the presence of the amide functionalities. Table 2 summarizes the chemical composition of the reference PES and the modified membranes. The untreated PES membrane is composed of 71.6% carbon, 24.4% oxygen, and 3.9% sulfur.

The application of the thin amide layer changed the composition measured at the membrane surface, and a significant increase in nitrogen on the surface of the membrane was detected. Since the reference PES membrane does not contain any nitrogen, this effect indicates that amide functionalities were formed on the membrane surface, i.e., the modification was successful. The formation of the amide coating can be further proved by the C1s spectra (Figure 5). Three signals were observed for the reference and modified PES membranes: a major peak at 285 eV that corresponds to carbon atom without adjacent electron-withdrawing atoms (C−C and C−H), an intermediate peak at 286.5 eV which is assignable to a carbon in weak electron-withdrawing atoms (C−O−C), and a minor peak at 288.5 eV which is associated with carbons attached to strong electron-withdrawing atoms (carboxylic O=C−O and amides O=C−N) [46].

#### 3.1.5. E2 Adsorption

The adsorption properties of the untreated and the modified PES membranes were examined by conducting batch adsorption tests for removal of E2 from aqueous solution. The effect of various synthesis parameters including types of monomers (HMD or PIP and TMC or ADC), irradiation dosages (150 or 200 kGy), and toluene as the swelling agent on the adsorption performance of the membranes were studied. Membrane disks (10 mm) were placed in a 48-well microtiter plate. Two hundred microliters of the E2 stock solution with an initial concentration of 5 mg∙L^−1^ was added to each membrane disk. The depletion of the E2 concentration was evaluated after 30 min. Figure 6 shows the results of adsorption capacities calculated by Equation (2).

PA-6 and PA-10 exhibit the highest adsorption capacities toward E2. Please note that in the case of PA-6 and PA-10 the amide functionalities were formed on the surface of the PES membranes with the interfacial polymerization reaction between either PIP and ADC or HMD and TMC, respectively. In 30 min, both PA-6 and PA-10 remove more than 60% of E2 present in the solution, while only slightly more than 30% was removed with the untreated PES membrane. These high enhancements in the adsorption capacities of PA-6 and PA-10 for E2 may indicate that hydrogen bonds between the hydroxyl group of E2 and the amide functional groups on the modified membranes were formed. In addition, the comparable enhancements in adsorption capacities of PA-6 and PA-10 probably indicate that the formation of hydrogen bonds is regardless of the aromatic or aliphatic character of the amide functional group created on the PA-6 and PA-10. On the other hand, a 20% difference in E2 adsorption capacity is observed by comparing adsorption performances of PA-10 and PA-9. No toluene was added in the process of modification in the case of PA-9. These results reveal the key role of toluene in the high adsorption performance of the modified membranes. The same behavior is observed for all the modified membranes, confirming the mentioned finding on the important effect of toluene on adsorption capacities of the modified membranes for E2. The higher adsorption capacity by adding toluene may be attributed to the swelling of the membrane. In fact, a swelling-driven effect of toluene can result in an increase in the surface area of the membranes, i.e., a higher concentration of amide groups is accessible. It is worth noting that only soaking an untreated PES membrane in toluene was not sufficient for increasing the adsorption performance of the membrane for E2. Therefore, toluene plays an important role in the amide modification by swelling the membrane. The E2 adsorption results also revealed that lower electron beam irradiation dosage is more successful to immobilize the amine monomer on the surface of the PES membrane. The average E2 adsorption capacity measured for all the modified membranes was 0.58 µg cm^−2^ (mass adsorbed per unit membrane area), which is nearly a two-fold increase compared to Koyuncu et al. [47], who reported an E2 adsorption capacity of 0.34 µg cm^−2^ on a polyamide thin film composite nanofiltration membrane (NF200). PA-6 and PA-10 with adsorption capacity of 0.82 µg cm^−2^ had the highest E2 adsorption capacity. This value is slightly higher than the maximum adsorption capacity (78 µg cm^−2^) of an ultrafiltration PES membrane for E2 which was reported by Jermann et al. [48]. The results here are comparable with the work of Han et al. [34] who demonstrated that the high adsorption capacity originates from the hydrogen bonding between PA amide groups and proton-donating moieties on E2 molecules. Values of E2 adsorption [%] on Ref and modified PES membranes are presented in Appendix A.

## 4. Conclusions

This work demonstrates the efficient removal of E2 from water by PES microfiltration membranes modified with amide functional groups. In fact, the microfiltration PES membrane surfaces were successfully modified with an amide functional coating. The modified membranes showed a high E2 adsorption capacity. Interestingly, membrane surface modification by both alkyl and aromatic amide functionalities resulted in comparable E2 adsorption properties. We, therefore, conclude that hydrophobic interactions were not significantly involved in the adsorption process. It can rather be discussed that the successful formation of hydrogen bonds between E2 and amide coating is responsible for such high adsorption capacities of the modified membranes toward E2. The modified membranes also had a slightly higher water wettability and water permeance compared to those of the untreated PES membranes. The pore structure on the other hand was not changed which indicates a very thin or even monomolecular layer of the amide modification.

The effects of synthetic parameters on the modified membranes were also studied and compared. Adding toluene was found to have the strongest effect on creating amide functional groups on the surface of the PES membrane with an adsorption capacity of 0.82 µg cm^−2^ probably by swelling the membrane.

The present study clarifies that the surface modification by amide functionalities is an efficient and inexpensive method to generate stable and high-performance adsorber membranes. In contrast to traditional PA thin-film composite membranes, the amide coated membranes retain and, in some cases, even improve their original microfiltration permeation performances.

## Figures and Tables

**Figure 1 membranes-11-00099-f001:**
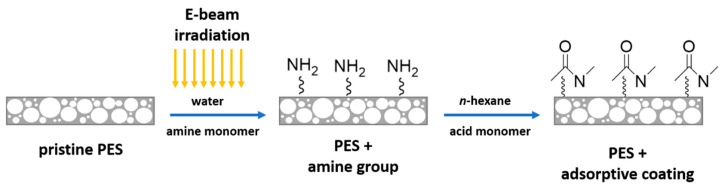
Schematic illustration of the interfacial reaction on the surface of the polyethersulfone (PES) membrane.

**Figure 2 membranes-11-00099-f002:**
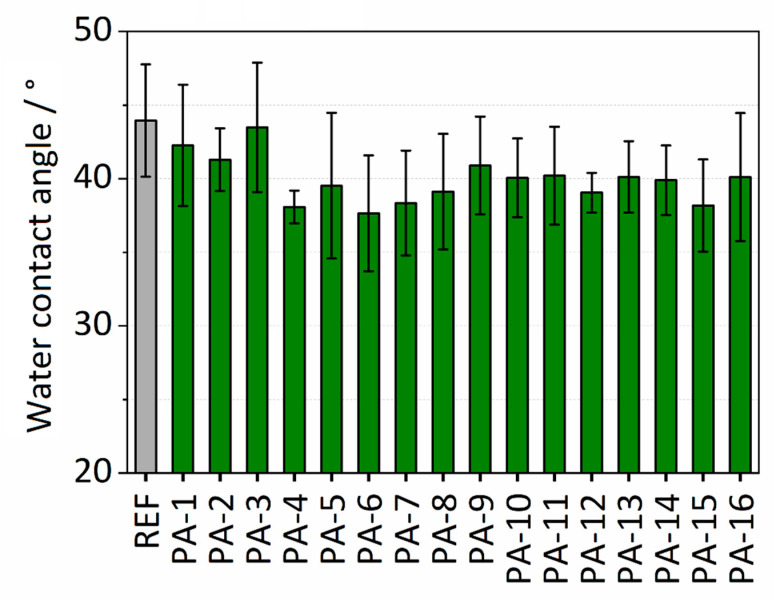
Water contact angles of modified and untreated (REF) polyethersulfone (PES) membranes.

**Figure 3 membranes-11-00099-f003:**
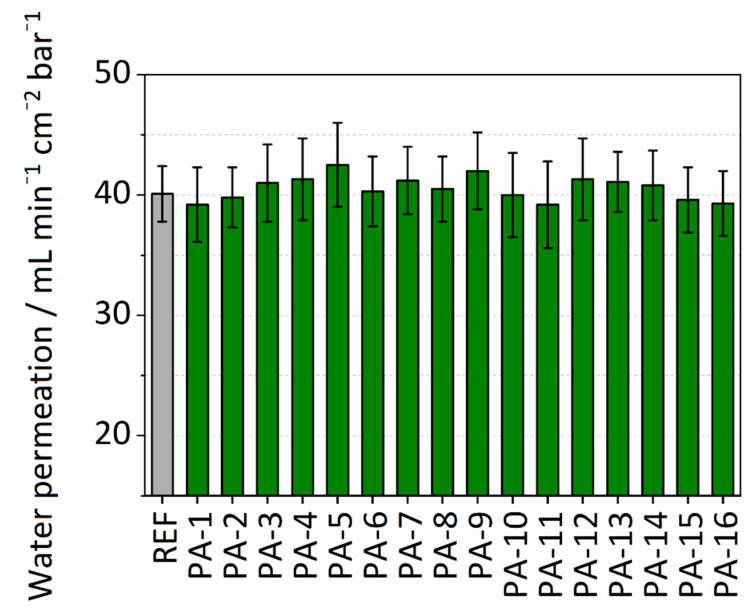
Water permeability of modified and untreated PES membranes.

**Figure 4 membranes-11-00099-f004:**
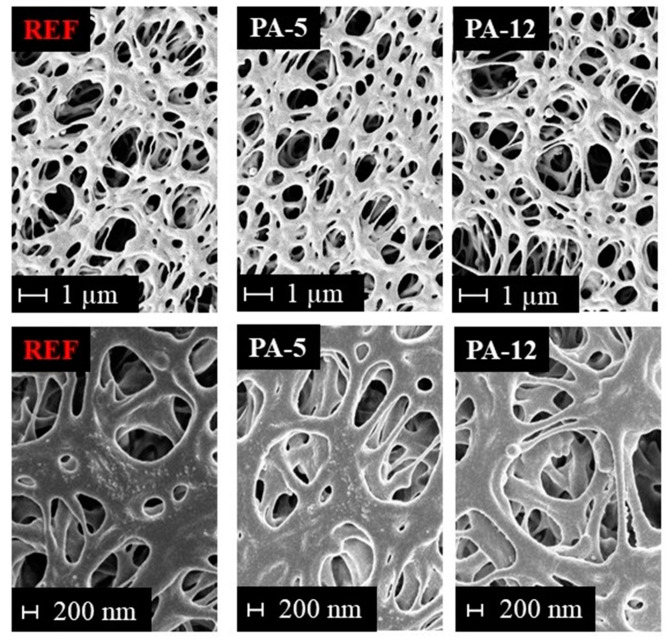
SEM pictures of untreated and modified membranes at different magnifications: 10,000-fold (top) and 25,000-fold (bottom).

**Figure 5 membranes-11-00099-f005:**
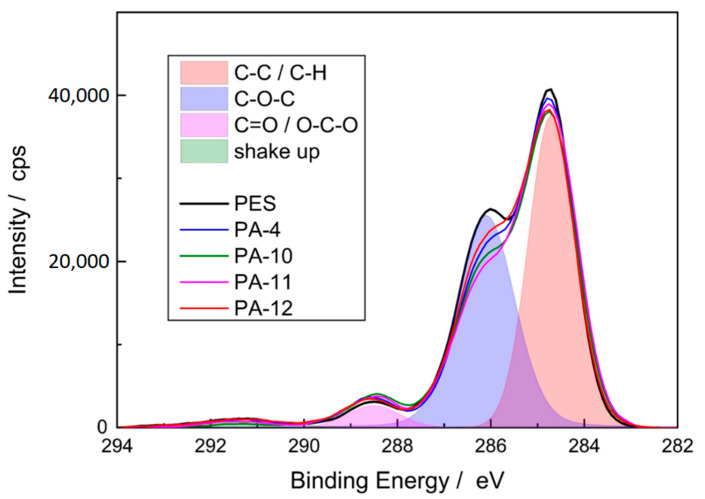
C1s spectra of reference and modified PES membranes. The overlap region is related to reference PES membrane.

**Figure 6 membranes-11-00099-f006:**
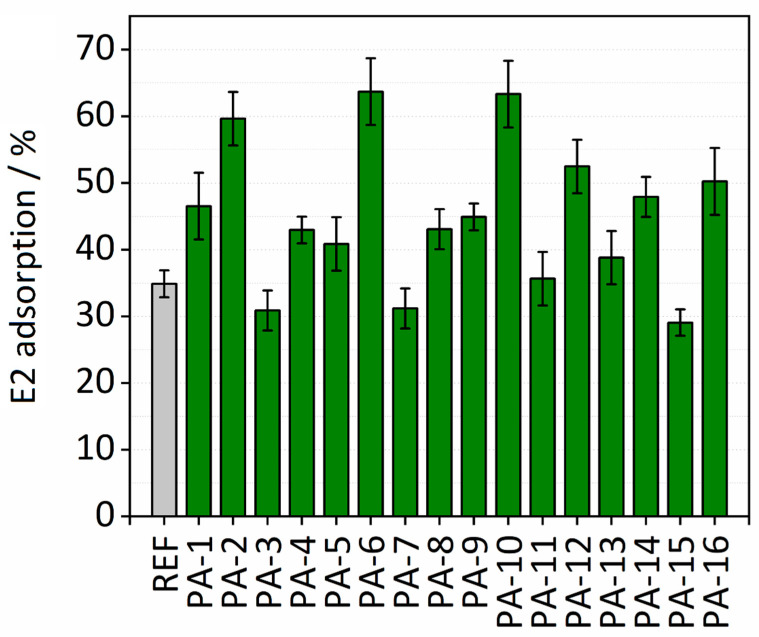
E2 adsorption (%) on modified and untreated PES membranes.

**Table 1 membranes-11-00099-t001:** Modification parameters.

	Amine Monomer 2 wt.%, 30 min	Acid Monomer 0.2 wt.%, 2 min	Radiation Dose (kGy)	Toluene
PA ^1^-1	PIP ^2^	TMC ^3^	150	-
PA-2	PIP	TMC	150	10 mL
PA-3	PIP	TMC	200	-
PA-4	PIP	TMC	200	10 mL
PA-5	PIP	ADC	150	-
PA-6	PIP	ADC	150	10 mL
PA-7	PIP	ADC	200	-
PA-8	PIP	ADC	200	10 mL
PA-9	HMD	TMC	150	-
PA-10	HMD	TMC	150	10 mL
PA-11	HMD	TMC	200	-
PA-12	HMD	TMC	200	10 mL
PA-13	HMD	ADC	150	-
PA-14	HMD	ADC	150	10 mL
PA-15	HMD	ADC	200	-
PA-16	HMD	ADC	200	10 mL

^1^ Modified membranes (PA-1 to PA-16); ^2^ Piperazine; ^3^ Trimesoyl chloride.

**Table 2 membranes-11-00099-t002:** Chemical composition of the untreated and modified membranes determined by X-ray photoelectron spectroscopy (XPS).

	Chemical Composition (Relative Atom %)			
Label	C	N	O	S
REF ^1^	71.6	-	24.4	3.9
PA-1	71.8	0.3	23.9	3.8
PA-2	71.5	0.3	24.2	3.9
PA-3	70.5	0.2	25.4	3.8
PA-4	71.3	0.1	24.6	3.9
PA-5	71.7	0.2	24.3	3.7
PA-6	69.9	0.3	25.9	3.8
PA-7	70.6	0.3	25.5	3.6
PA-8	71.0	0.3	24.9	3.7
PA-9	71.4	0.2	24.5	3.8
PA-10	69.9	0.2	26,0	3.9
PA-11	70.9	0.2	24.8	3.9
PA-12	70.1	0.2	25.7	3.9
PA-13	69.9	0.2	26.0	3.8
PA-14	70.6	0.2	25.3	3.8
PA-15	70.9	0.1	25.1	3.8
PA-16	71.23	0.1	24.7	3.8

^1^ untreated polyethersulfone membrane.

## Data Availability

Not applicable.

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
