# Peer review of "Estradiol Removal by Adsorptive Coating of a Microfiltration Membrane"

_membranes, 2021, doi:10.3390/membranes11020099_

Round 1

Reviewer 1 Report

Comments for authors:

  1. The manuscript should include conventional layout, eg. introduction, materials, methods, results conclutions and references or introduction results, concultions, materials and methods and references. In draft you mixed above styles, please choose one.
  2. The XPS analysis sholud be presented at graph and showed specific region related with chemical bonds. 
  3. It is hard to read and understand. Authors cannot mixed the material analysis like contact angles with water permeace. Authors should work on it and present results more clear.

Author Response

Thank you very much for considering our manuscript for publication in Membranes. We appreciate your thorough examination of the manuscript, and we carefully revised the manuscript for obtaining now a better quality of the paper. Therefore, we would like to resubmit our manuscript and hope we were able eliminate all mistakes and misunderstandings according to your suggestions.

We responded to the different review comments as follows:

  • The manuscript should include conventional layout, eg. introduction, materials, methods, results conclutions and references or introduction results, concultions, materials and methods and references. In draft you mixed above styles, please choose one.

Many thanks for this comment! The layout was corrected. The sequence now is: introduction, materials and methods, results and discussion, conclusion, references.

  • The XPS analysis sholud be presented at graph and showed specific region related with chemical bonds.

The C1s spectra of the untreated and modified PES membranes were included. (Figure 5). All the other modifications had slightly similar chemical bonds and composition as stated in table 2.

  • It is hard to read and understand. Authors cannot mixed the material analysis like contact angles with water permeace. Authors should work on it and present results more clear.

A schematic illustration was added to clarify the idea of the manuscript (Figure 1). Contact angle analysis and water permeance were reported in two different sections. It was noted that the decreased water contact angle (hence the wettability) influenced the permeance of the membranes (lines 257-261).

Reviewer 2 Report

Journal: Membranes (ISSN 2077-0375) 

Manuscript ID: membranes-1062656

Title: Estradiol Removal by Adsorptive Coating of a Microfiltration Membrane

This manuscript focuses on the effect of the enhancement of the adsorption properties of polyethersul-fone microfiltration membranes for 17β-estradiol.After carefully examining the manuscript, I think this manuscript is good in terms of ideas, language and structure, but there are still some problems.So I suggest that the manuscript can be accepted after revision.

My comments are as follows:

  • In 2.1 Water contact angle, please explain the specific reasons for the change in water contact angle.
  • In Water permeance,why does the increased wettability of the membrane surface increase the water permeability? Please explain the reason in detail.
  • Authorsdescribed in the manuscript,“The SEM results also revealed that this amide modification on the PES membrane is very thin and cannot be detected using SEM”.Is there any other method to characterize its thickness? 
  • In Membrane chemical composition,in addition to XPS, energy dispersive X-ray spectroscopy can also be used to make it more intuitive to indicate the distribution and quantity of its elements.
  • In this manuscript,authorsrepeatedly mentioned that the modified membrane has good robustness, but there is no data to support it.

Author Response

Thank you very much for considering our manuscript for publication in Membranes. We appreciate your thorough examination of the manuscript, and we carefully revised the manuscript for obtaining now a better quality of the paper. Therefore, we would like to resubmit our manuscript and hope we were able eliminate all mistakes and misunderstandings according to your suggestions.

We responded to the different review comments as follows:

This manuscript focuses on the effect of the enhancement of the adsorption properties of polyethersul-fone microfiltration membranes for 17β-estradiol.After carefully examining the manuscript, I think this manuscript is good in terms of ideas, language and structure, but there are still some problems.So I suggest that the manuscript can be accepted after revision.

  • In 2.1 Water contact angle, please explain the specific reasons for the change in water contact angle.

The reason for the change in water contact angle was added (lines 238-243).

  • In Water permeance,why does the increased wettability of the membrane surface increase the water permeability? Please explain the reason in detail.

The reason for the increased permeance of the membrane with the increase in wettability was explained in lines 257-261.

  • Authorsdescribed in the manuscript,“The SEM results also revealed that this amide modification on the PES membrane is very thin and cannot be detected using SEM”.Is there any other method to characterize its thickness?

It is not possible to analyze the thickness of this PA layer. The PES membrane is a porous system and not a flat system. The PA modification is a very thin molecular layer which consists of the same type of atoms as the base membrane. It is challenging to differentiate between the PA modification layer and the polymer membrane below. The previously published work also described the challenges in the analysis of the thickness of modification layer. See below: (Breite, Went et al. 2020).

  • In Membrane chemical composition,in addition to XPS, energy dispersive X-ray spectroscopy can also be used to make it more intuitive to indicate the distribution and quantity of its elements.

EDX is unsuitable for lower elements (C, N, O), and the PES membranes are mainly composed of these elements.

  • In this manuscript,authorsrepeatedly mentioned that the modified membrane has good robustness, but there is no data to support it.

The modified membranes have been investigated in previous work in terms of stability, see below: (Breite, Went et al. 2020). We assume that this modification is a similar system, and the modified membranes are stable. The analysis of the robustness of this modification shall be part of future studies. Therefore, the phrase “robust” has been replaced by “stable” (line 26 and line 374).

Reference:

Breite, D., et al. (2020). "Reduction of Biofouling of a Microfiltration Membrane Using Amide Functionalities—Hydrophilization without Changes in Morphology." Polymers 12(6): 1379.

Reviewer 3 Report

The manuscript reports functionalization of PES microfiltration membrane through interfacial reactions. The key parameters for the functionalization are optimized and good filtration performance is demonstrated. However, a critical flaw of the manuscript is that the origin of enhanced performance is not fully clarified. The authors prepared a number of modified membrane and the critical properties of these membranes (water contact angle, water permeation and surface morphology) are quite similar. The difference in the filtration performance among these membranes are not reasoned with the current results. The authors should provide concrete experimental evidence to clarify the key properties of the membrane, which are closely related to the filtration performance. Besides, the introduction section is too lengthy; the language of the manuscript can be further polished.

Author Response

Thank you very much for considering our manuscript for publication in Membranes. We appreciate your thorough examination of the manuscript, and we carefully revised the manuscript for obtaining now a better quality of the paper. Therefore, we would like to resubmit our manuscript and hope we were able to eliminate all mistakes and misunderstandings according to your suggestions.

We responded to the different review comments as follows:

  • The manuscript reports functionalization of PES microfiltration membrane through interfacial reactions. The key parameters for the functionalization are optimized and good filtration performance is demonstrated. However, a critical flaw of the manuscript is that the origin of enhanced performance is not fully clarified.

The reason for the enhanced performance of the membranes regarding wettability is included in lines 238-243. In the case of water permeation enhancement, the reason is included in lines 257-261. And the origin of the enhanced performance of the modified membranes in terms of increased estradiol adsorption was thoroughly explained in the lines 325-330.

  • The authors prepared a number of modified membrane and the critical properties of these membranes (water contact angle, water permeation and surface morphology) are quite similar. The difference in the filtration performance among these membranes are not reasoned with the current results. The authors should provide concrete experimental evidence to clarify the key properties of the membrane, which are closely related to the filtration performance.

The focus of the manuscript is the enhancement in the adsorption properties (not filtration performance!). Static adsorption tests were carried out as described in lines 204-205 and 212-215. Adsorption tests during filtration will be part of future studies.

  • Besides, the introduction section is too lengthy.

The concept of the modification needed a comprehensive discussion on the origin of the increase in the adsorption capacity. Authors believe no part of the introduction can be omitted.

  • the language of the manuscript can be further polished.

The manuscript was examined thoroughly, and the errors were corrected.

Round 2

Reviewer 1 Report

I can accept this manuscript as it is. 

Reviewer 3 Report

The authors have addressed some of my concerns. I recommend acceptance of the manuscript.